# Comparison of the Effectiveness of Radiotherapy with 3D-CRT, IMRT, VMAT and PT for Newly Diagnosed Glioblastoma: A Bayesian Network Meta-Analysis

**DOI:** 10.3390/cancers15235698

**Published:** 2023-12-03

**Authors:** Shan Xu, Rezarta Frakulli, Yilan Lin

**Affiliations:** 1Department of Ophthalmology, University Hospital Essen, 45147 Essen, Germany; shan.xu@uk-essen.de; 2Department of Oncology, Mianyang Central Hospital, Mianyang 621000, China; 3Department of Particle Therapy, University Hospital Essen, West German Proton Therapy Centre Essen, West German Cancer Centre, 45147 Essen, Germany; rezarta.frakulli@uk-essen.de

**Keywords:** glioblastoma, radiotherapy, proton therapy, meta-analysis, GBM

## Abstract

**Simple Summary:**

This study investigated the effectiveness of different modern radiotherapy strategies in the treatment of patients with newly diagnosed glioblastoma using a comprehensive review of the literature from sources such as MEDLINE, Embase and the Cochrane Central Registry. It included randomized and nonrandomized trials on treatments such as three-dimensional conformal radiotherapy, intensity-modulated radiotherapy and proton therapy, assessing their impact on overall survival and progression-free survival. A Bayesian network meta-analysis was utilized to provide a nuanced comparison of the varied therapies. The analysis of 816 patients from six studies indicated the significant benefit of proton therapy in terms of the survival outcomes, suggesting that proton therapy might be most effective for primary glioblastoma.

**Abstract:**

Background: This study aimed to assess the relative efficacy of modern radiotherapy strategies in patients with newly diagnosed glioblastoma. Method: A comprehensive literature review was conducted through MEDLINE, Embase and the Cochrane Central Registry of Controlled Trials of studies focused on newly diagnosed glioblastoma published up to and counting 15 September 2022. We included randomized controlled trials (RCTs) and comparative nonrandomized studies (NRSs) of radiotherapy for newly diagnosed glioblastoma. Eligible studies included patients treated with three-dimensional conformal radiation therapy, intensity-modulated radiation therapy, volumetric modulated arc therapy or proton therapy reporting either overall survival, progression-free survival or both. The impact of different radiotherapy modalities on survival was evaluated by direct comparisons of indirect evidence and estimated hazard ratios in terms of a Bayesian network meta-analysis. Results: A total of six RCTs or NRSs comprising 816 glioblastoma patients with modern radiotherapy strategies were reviewed, yielding improved overall survival by proton therapy over all other regimens. The network meta-analysis also indicated a significant advantage of proton therapy compared with other radiotherapy strategies in regard to progression-free survival. Conclusion: Our findings suggested PT as a standard RT regime with possibly superior survival outcomes for selected patients with GBM.

## 1. Background

Glioblastoma (GBM), World Health Organization grade IV, is the most frequent malignancy of the central nervous system (CNS) in adulthood, accounting for 45% to 50% of all gliomas [1]. Despite growing research on tumor biology and molecular genetics in the recent years [2], GBM patients usually show a fast and aggressive course with a median survival of less than one year. The treatment of choice for newly diagnosed GBM patients is gross total resection followed by postoperative radiotherapy (RT) and temozolomide (TMZ) chemotherapy [3], whereas the amendment of lomustine as per the CeTeG/NOA-09 trial might improve the survival of patients with methylation of MGMT promoter [4]. With respect to RT, 60 Gy in 30 fractions is considered the standard dose maximum to avoid radiation-induced brain injury [5]. Recent advancements in RT technology such as intensity-modulated radiation therapy (IMRT), volumetric modulated arc therapy (VMAT) and proton therapy (PT) enable the delivery of a higher RT dose to the tumor target while sparing the surrounding healthy brain tissue with increasing dose conformity [6,7]. GBM may occur at diverse intracerebral sites, probably close to critical organs at risk (OAR) such as the brainstem, optic nerves and chiasma, causing severe late RT toxicities, in particular, radiation necrosis and neurocognitive deficits. Thus, it is important to use the most optimal RT technique to ensure the maximum target volume coverage while reducing the RT dose to critical OAR simultaneously in order to improve the survival outcomes with acceptable toxicity.

Several studies have suggested that high-dose radiotherapy (RT) might enhance survival and local control in GBM patients [8,9]. However, it remains unclear whether increasing the RT dosage could improve survival outcomes without adversely affecting their quality of life. PT, with its favorable characteristics of dose distribution such as a spread-out Bragg peak, reduces the entrance dose, and the absence of an exit dose offers the potential for dose escalation [7]. Such an approach could enhance the dose coverage and conformity, especially for challenging target volumes next to critical structures [10]. Moreover, limited radiation exposure to normal tissues outside the target area by means of PT might mitigate treatment-related toxicity rates [11]. Notably, the maintenance of quality of life has become an essential part of comprehensive cancer care. Since clinicians and patients may be confused by diverse modern RT modalities for GBM with respect to their advantages and disadvantages, we designed this network meta-analysis to help the decision making.

## 2. Method

### 2.1. Literature Search

Literature screening was performed according to the method outlined in the Cochrane Handbook for Systematic Reviews of Intervention [12] and Meta-analyses (PRISMA) reporting guideline (in the Appendix A). Institutional review board approval was not required. We conducted a comprehensive literature search of electronic databases including MEDLINE, EMBASE and the Cochrane Central Register of Controlled Trials (Central) databases from inception up to and including 15 September 2022. The search strategy was based on a combination of subject headings and keywords related to the concept of “Irradiation”, “Radiotherapy”, “Intensity-modulated radiotherapy”, “Volumetric modulated arc therapy”, “Three-dimensional conformal radiotherapy”, “Proton therapy”, “IMRT”, “VMAT”, “3DCRT”, “Glioblastoma”, “Glioblastoma multiforme”, “GBM”, “High-grade glioma” and “HGG”. Moreover, the references of the selected articles and reviews were manually retrieved to obtain all potentially relevant studies. Retrieved articles were screened and reviewed for their eligibility by two independent reviewers (SX, RF). Differences in the determination of a study’s eligibility were resolved by consensus or discussed with a third person as referee (YLL). The language of publications was restricted to English.

### 2.2. Study Selection

We incorporated randomized controlled trials (RCTs), quasi-randomized trials, nonrandomized studies (NRSs) and controlled before-and-after studies, all with pertinent concurrent comparison groups. The eligibility criteria for studies included in our meta-analysis were: (1) patients pathologically diagnosed with newly onset glioblastoma multiforme (GBM) without any prior treatment; (2) treatment involving modern radiotherapy (RT) modalities such as 3D conformal radiotherapy (3DCRT), intensity-modulated radiotherapy (IMRT), volumetric modulated arc therapy (VMAT) or proton therapy (PT); (3) an emphasis on overall survival (OS) and progression-free survival (PFS) as the primary outcomes; and (4) a minimum average follow-up duration of one year. Studies meeting the following criteria were excluded: (1) Types of publications being abstracts, conference papers or reviews. (2) Insufficient or inadequate data provision (patient count less than 10).

### 2.3. Data Extraction and Risk of Bias Assessment

Data extraction from each study was independently implemented by two reviewers, SX and RF, utilizing a standardized, predesigned form in Microsoft Excel. Extracted data included: the first author’s name, publication year, patient demographics, treatment strategies, sample size, number of patients assessed for response, treatment dosage and schedule, median number of temozolomide cycles administered and outcomes (median OS and median PFS). When available, hazard ratios (HRs) for PFS and OS were extracted along with their 95% confidence intervals (CIs). In cases where HRs and corresponding CIs were not explicitly reported, we estimated them by reconstructing individual patient data from published Kaplan–Meier curves, following the method described by Guyot et al. [13]. Authors of the included studies were contacted for clarification or additional information if essential data were missing or unclear. The risk of bias in randomized trials was independently evaluated by reviewers SX and YLL using the Cochrane Collaboration’s tool and the risk of bias (RoB 2.0) tool [14]. Any discrepancies between reviewers were resolved by reaching a consensus.

### 2.4. Data Synthesis and Analysis

This study was administered in accordance with the Preferred Reporting Items for Systematic Reviews and Meta-Analyses (PRISMA) extension statement, specifically for systematic reviews [15]. As all analyses were performed on data from previously published trials, it was not necessary to obtain ethical approval or patient consent. The methodology for the Bayesian network meta-analysis followed the established procedures described previously [16,17]. In this work, we conducted a comprehensive synthesis of evidence focusing on two primary outcomes: progression-free survival (PFS) and overall survival (OS). For each of these outcomes, a Bayesian network meta-analysis was constructed utilizing the Markov Chain Monte Carlo (MCMC) simulation technique. This involved 100,000 iterations across three separate chains to ensure the robustness and reliability of the results. We chose noninformative priors, specifically normal distributions with a mean of 0 and a large variance (N[0, 10,000]) as our effect parameters to maintain objectivity and minimize bias in the analysis. A key component of our analysis was the creation of a network plot. This visual representation illustrated the evidence base, highlighting various treatment modalities as nodes interconnected by lines. The size of each node was proportionate to the number of patients receiving the respective radiation treatment, thus providing immediate insight into the extent of evidence for each modality. The interconnecting lines varied in thickness, indicating the strength and directness of the relationships between treatments. This thickness was not arbitrary but was systematically weighted based on the amount of direct evidence available for each treatment comparison. This comprehensive approach allowed for a nuanced understanding of the treatment landscape for PFS and OS in the context of the present evidence.

Moreover, we executed an analysis using the fixed-effects model because only one trial provided direct evidence for most of the treatment comparisons. Nevertheless, a random-effects model was introduced, while sensitivity analysis and model fits were reviewed in line with the Deviation Information Criteria [18]. In the differentiation of any two models, if the Deviation Information Criteria of one model is at least 5 less than that of another model, it can be considered a better fit model. Network heterogeneity was assessed using the Cochrane Q (χ2) test and quantified using the I2 statistic within each pairwise comparison. Herein, two or more studies were available for distinction [19], and the values of *I^2^* statistics were 25%, 50% and 75% indicating mild, moderate and high heterogeneity, respectively. In our network, it was unusual to have both direct and indirect evidence for most of the comparisons. We therefore considered our analysis to be consistent, i.e., direct and indirect evidence, when both available types of evidence for a given comparison were statistically similar. To test the robustness of this assumption, the node-splitting method was used to assess incoherence in any closed loops [18,19]. Relative effects of treatments were reported as HRs for survival outcomes (PFS and OS) along with corresponding 95% credible intervals (CIs), the Bayesian equivalent of 95% CIs. Rank probabilities were also calculated in terms of the hierarchy of each treatment and the matrix and plot of rank probabilities provided by the gemtc package. The network meta-analysis was performed in WinBUGS software (version 1.4.3, MRC Biostatistics Unit) using R software (Version 4.2.3). 

## 3. Results

### 3.1. Overall Characteristics of Selected Studies and Quality of Evidence

Based on the above-described search strategy, the literature search yielded 335 records. Of these, 155 were excluded after screening the titles and abstracts because of duplicates (Appendix A). The full texts of the remaining 180 articles were evaluated by the reviewers with 174 studies excluded due to them being reviews, letters, case reports or uncontrolled trials. Finally, a total of six articles including 816 GBM patients were enrolled. The characteristics of the RCTs and NRSs included in this meta-analysis are summarized in Table 1. The study sample sizes ranged from 18 to 174, and studies were published between 2013 and 2021. The risk of bias and the quality assessment in all studies are presented in Appendix A indicating the quality of the included studies. In addition, according to the MCMC model, *I^2^* was estimated to be 0.00%. Therefore, there was no heterogeneity in the data, and our results were stable and reliable.

### 3.2. Overall Survival

Six trials (816 patients) comparing 3DCRT, IMRT, VMAT and PT were included in the OS analysis (Figure 1A). The efficacy of different treatments in terms of the HR and corresponding 95% CI are displayed in Figure 2A. As indicated in the result, PT was the most likely regimen to exhibit a higher OS compared with other strategies. The key comparison treatments included PT vs. VMAT with an HR of 1.03 (95% CI, 0.53–2.01), and PT vs. IMRT with an HR of 1.16 (95% CI, 0.64–2.13). Figure 3A illustrates a direct plot of rank probabilities, from which we could easily find the ranking of each RT regimen. Similar results showed that PT had the highest probability of being associated with the best OS. Figure 3B is a cumulative rank plot with the surface under the cumulative ranking curve (SUCRA) of each intervention. The detailed values are presented in Table 2. Consistently, the SUCRA analysis suggested that PT demonstrated the highest probability of being associated with the best OS (SUCRA: 72.6%), followed by VMAT (SUCRA: 66.5%) and IMRT (SUCRA:44.9), whereas 3DCRT was least likely to be the optimal treatment strategy regarding OS (SUCRA: 26.3%). The node-splitting method and its relative the Bayesian *p*-value were utilized to report the inconsistency of our results. The Bayesian *p*-values of the node-splitting method were >0.05, meaning the direct and indirect evidence was consistent (Appendix A).

### 3.3. Progression-Free Survival

Four trials of 550 patients comparing 3DCRT, IMRT, VMAT and PT were included in the PFS analysis (Figure 1B). Figure 2B shows the HRs and corresponding 95% CIs of each RT strategy. PT was again the most likely regimen yielding a higher PFS compared with other treatment methods. The direct plot of the rank probabilities results also confirmed PT as having the highest probability of being associated with the best PFS (Figure 3A). The estimated SUCRA values were 78.25% for PT, followed by IMRT with 57% (Figure 3B and Table 2), suggesting that both strategies were associated with the highest likelihood for the improved PFS of patients with GBM. We used the node-splitting method and its relative the Bayesian *p*-value to report the inconsistency of our results. Bayesian *p*-values of the node-splitting method of >0.05 (Appendix A) demonstrated the consistency of the direct and indirect evidence.

## 4. Discussion

RT technology has advanced dramatically over the last few decades [25,26]. In contrast to traditional 2D treatment strategies, 3DCRT has greatly improved dose distribution and, subsequently, local control for prostate cancer and other tumor entities [6,26,27,28,29]. To date, 3DCRT has been the most available RT planning modality worldwide. Three-dimensional CRT was the initial approach, whereas the radiation beams were adapted to fit the shape of the tumor. Based on computed tomography (CT), 3DCRT creates a three-dimensional illustration of the tumor and surrounding normal structures, allowing photon beams to be more precisely aimed at the tumor from different angles as well as reducing the exposure of and damage to adjacent healthy tissue [30]. However, the oncological outcomes remained unsatisfactory. Unlike 3DCRT, IMRT significantly decreased the average dose discharged to critical OAR such as the optic chiasm, brainstem, optic nerves and healthy brain regions while improving dose homogeneity and target volume coverage [31]. IMRT is an advanced form of 3D radiation therapy using computer-controlled linear accelerators (LINACs) for the accurate dose delivery of a radiation treatment to a predefined tumor site. The radiation isodose is designed to match the three-dimensional shape of the target by modulating the beam intensity in multiple small volumes. In this way, IMRT enables higher-dose radiation to be focused on the target volume while minimizing the risk of an unnecessary dose to adjoining normal tissue and OAR [32]. With respect to 3DCRT, IMRT is able to treat a gross tumor surrounded by critical structures with a higher prescription dose (i.e., 70 Gy) in compliance with the dose constraints of the respective OAR [33]. Wagner et al. [34] and Thilmann et al. [35] illustrated that IMRT had a higher target volume coverage than 3DCRT with V95% improvements of 13.5% and 13.1%, respectively, especially when planning a target volume (PTV) close to critical OAR [34]. According to the dosimetric study of GBM patients of MacDonald et al. [36], IMRT noticeably enhanced tumor control (*p* < 0.005) while lowering dose delivery to the brainstem and brain (*p* < 0.033) compared with 3DCRT. In the clinical investigation of GBM treatments [20], there was a slight but not significant improvement between the IMRT and the 3DCRT groups for 1-year OS (89.6% vs. 75.8%) and 1-year PFS (61.0% vs. 45.5%). The authors concluded that IMRT might attain a better survival rate by also applying the hypofractionated regimen that was confirmed in recent studies [37,38]. IMRT in modestly hypofractionated schemes (2.4 – 3Gy per fraction) without a significant increase in total dose may improve OS and PFS with a standard TMZ schedule in GBM treatment. As per the dosimetric benefits and clinical outcomes, IMRT is generally considered superior than 3DCRT for the treatment of GBM. According to our analysis, we confirmed this result as well.

In contrast to IMRT, VMAT has analogous performance but a distinctly lower dose delivery time, increasing clinical efficiency and the quality of the treatment [39,40]. Generally speaking, VMAT is a type of specialized IMRT that provides photon beams by rotating the linear accelerator around the patient. Unlike conventional IMRT where the LINAC stops and starts to treat different parts of the tumor, VMAT delivers radiation continuously during a steady motion of the LINAC. Consequently, this technique allows for quicker treatments, oftentimes in a single rotation or fewer rotations around the body, reducing the total treatment time with potentially increased precision [41]. In addition, VMAT has many advantages over IMRT, including better target coverage and dose conformity while reducing the dose to the optic chiasm, brainstem, cochlea and hippocampi [42,43]. Thus, VMAT may achieve similar local control with diminished RT-related toxicity. Navarria et al. evaluated 341 patients (74% with GBM) undergoing either VMAT or 3DCRT with comparable PTV coverage [21]. Herein, VMAT showed markedly improved OS (19 months) and PFS (15 months) compared with 3DCRT (15 and 12 months, respectively) in a median follow-up period of 1.3 years. Moreover, recurrence was reduced by 10% in cases receiving VMAT. In terms of the comparison of IMRT with VMAT among patients with frontal and temporal high-grade glioma (HGG), VMAT considerably decreased the maximum and average doses to the retina, contralateral optic nerves and lens (*p* < 0.05), while doses to the chiasm, ipsilateral optic nerves and brainstem were similar [43]. According to the review of Sheu et al. [24], the survival and toxicity outcomes were comparable, whereas the mean time of treatment was significantly decreased by 29% with VMAT (10.3 min) compared to that of IMRT (14.6 min). In our analysis, VMAT was better than 3DCRT and IMRT, with a higher OS, but not regarding PFS. Based on these published data, VMAT has a clear advantage over 3DCRT in terms of a preferable PTV coverage, dose conformity and sparing of critical OAR. Furthermore, VMAT overcomes the negative aspect of IMRT in terms of its shorter beam delivery time with adequate dose conformity. It is increasingly used in the treatment of GBM because of its brief treatment time and excellent dosimetry in relation to IMRT. This result has also been replicated in the treatment of other tumors such as esophageal cancer [44] and locally advanced non-small cell lung cancer [45]. Notwithstanding these results, VMAT is significantly favorable with regard to practical and radiobiological aspects due to its significant reduction in treatment time in GBM. Moreover, VMAT creates less of a financial burden for the health care system. Corresponding to the 2017 Medicare fee schedule in their geographic area, the 2-arc VMAT delivered in 30 fractions resulted in a substantially lower cost of USD 9100 compared to the cost of IMRT of about USD 27,200 delivered with six noncoplanar gantry angles [24].

On the other hand, proton therapy is based on positively charged particles, i.e., protons, instead of photons from which other RT modalities (3DCRT, IMRT, VMAT) are derived. Compared with the conventional RT technique using photon beams, PT can target a tumor more accurately thanks to it physical advantages. Protons deposit the dose maximum at a predetermined depth in the tumor and then drops it almost perpendicularly with no exit dose, resulting in the optimal sparing of normal tissue and OAR behind the tumor. This makes PT particularly favorable for treating tumor sites close to critical structures such as optic nerves and the brainstem for GBM [46]. In the present study, the network meta-analysis identified significant survival benefits of PT over all the other RT strategies, providing high effectiveness in terms of enhancing PFS, followed by IMRT, VMAT and 3DCRT. Whereas VMAT and IMRT might be equally effective in terms of the optimal OS, 3DCRT might not have the best impact on prolonging OS and PFS. Our results were consistent with another analysis of proton vs. photon RT for primary gliomas as per the USA National Cancer Data Base [47]. GBM patients treated with PT yielded an increased 5-year survival (46.1% vs 35.5%, *p* = 0.0160) and mean survival (45.9 vs. 29.7 months, *p* = 0.009) compared with those receiving photon irradiation. The multivariate regression analysis illustrated similar findings in HGG (HR = 0.67, CI (0.53–0.84), *p* < 0.001) and low-grade glioma (LGG) patients (HR = 0.46, CI (0.22–0.98), *p* = 0.043) with a reduced mortality risk after PT. Brown et al. [1] also noted improved PFS (median 8.9 months in IMRT vs. 6.6 months in PT) and OS (median 21.2 months in IMRT vs. 24.5 months in PT), although not statistically significant. Interestingly, Rusthoven et al. [48] found better survival of GBM patients with RT-induced necrosis during the second surgery than those with no necrosis. Given the advantage of reducing critical OAR doses by PT, the benefit for primary GBM in light of survival and local control of an increased prescription dose needs to be investigated. Regarding less radiosensitive histologies, an encouraging recent large retrospective study of chondrosarcoma and chordoma showed that dose escalation and PT were associated with improved OS in patients with chondrosarcoma and chordoma [49].

As the role of the immune system in protecting against malignant disease is increasingly recognized, potential adverse effects of RT on the immune system due to myelosuppression need to be monitored, taking into account immune-sparing methods. In a randomized phase 2 trial, photon irradiation was compared with PT in the treatment of GBM with concurrent TMZ [1]. As a result, PT significantly reduced grade 3 lymphopenia. In contrast to high-grade disease, for LGG or benign CNS lesions, late and long-term sequelae such as neurocognitive impairments are the primary concern, taking into account their favorable prognostic outcomes with prolonged survival. In the dosimetric study of optic pathway glioma, PT was found to reduce nearly 50% of the maximum dose to contralateral ON as well as lower doses to the pituitary gland and chiasm [50]. Pituitary dysfunction and secondary hypopituitarism are cranial radiation-induced late adverse effects commonly found among long-time survivors [51,52]. In the present review of intracranial HGG cases at diverse sites, PT only significantly reduced the ON dose on the contralateral side. ON sparing is still relevant for HGG near the optic system in order not to compromise target volume coverage, observed at an increased minimum dose of CTV in PT compared to VMAT and IMRT [11]. 

The hippocampi and bilateral subventricular zone (SVZ) are known to harbor neural progenitor cells (NPCs) [53,54], contributing to neurogenesis and injury repair according to their self-renewing capacities [55,56], although the function of the stem cell niche’s dose remains unclear. For centrally localized craniopharyngioma, PT surpasses IMRT in terms of radiation dose exposure to the contralateral hippocampus and SVZ [57]. Notably, PT shows distinct dosimetric advantages for the hippocampi, entire brain, sensory organs and temporal lobes by preserving neuron activity following RT [57]. As a dose-dependent thinning effect on the cerebral cortex of 0.0033 mm/Gy was reported by Karunamuni et al. [58], pronounced in the temporal lobes and limbic cortex, long-term follow-up of HGG and LGG patients after RT indicated a frequent occurrence of dementia as well [59,60]. As the radiation-induced neurocognitive impairment rate is related to RT-exposed volume and critical OAR doses, PT may significantly reduce the risk for neurocognitive deficiency [61] that is essential for children, whose IQ, fine motor skills and processing speed are reduced following chemoradiation [62,63]. However, it is not substantial enough to prevent secondary malignancy in HGG patients because of their limited lifespan compared with patients of other intracranial tumor types [64,65].

Based on the aforementioned results, PT may be consider the most advantageous anti-GBM treatment because of its dose reduction in adjoining nontarget normal tissue without compromising target coverage, even for selected cases of unresectable or incompletely resected disease with a complex-shaped target near critical OAR. Although PT is expected to be comprehensively available worldwide, it is necessary to screen appropriate GBM patients for PT due to their generally restricted prognosis. Specifically, young patients and highly functioning adults with favorable clinical and molecular biological features may profit from PT owing to the improved quality of life and maintenance of neurocognitive functions. It is worth noting that, in addition to RT, the exploration of other adjuvants is ongoing. Although the current trials have not shown clear benefits yet, investigations of several strategies such as tumor treating fields and combined drugs [66,67] are underway. We believe that the diversity of treatment strategies across modes is promising, although this topic is not discussed in this paper. Should an effective pharmacological regimen emerge, it could significantly alter the conclusions we have currently drawn. 

There are limitations in the present network meta-analysis since many of the articles used in our analysis were retrospective. In addition, data from RCTs are still lacking and risk bias was not assessed in individual studies. However, certain retrospective articles can help to offer data that are unavailable from prospective trials with a small sample size. Furthermore, data reported in retrospective research may be collected in future prospective studies. In some studies, PT was supposed to be beneficial in sparing critical OAR without reducing doses to the target volume. Nonetheless, one should be cautious about these findings because the treatment plans were formulated at diverse institutes with distinct target delineation, treatment planning and prescription dose protocols. Moreover, target volumes were defined differently among the trials. Due to the above-mentioned limitations, more large-scale, well-designed studies should be conducted to report long-term follow-up effects offering more valuable data for clinical practice. 

## 5. Conclusions

Taken together, the network meta-analysis of the available retrospective data suggests that PT may provide the best clinical improvement for GBM compared to other modern RT modalities. The dosimetry advantage of PT is of clinical significance, including lower RT-induced toxicities and more favorable survival outcomes compared with photon irradiation. In addition, the clinical benefit of VMAT over 3DCRT is more evident than its benefit over IMRT. High-quality evidence from prospective randomized trials is needed to confirm these findings.

## Figures and Tables

**Figure 1 cancers-15-05698-f001:**
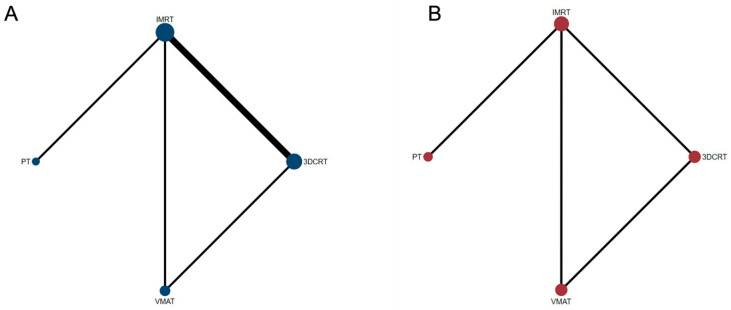
Network structure diagrams. (**A**) Overall survival (OS); (**B**) progression-free survival (PFS). In this network figure, each node represents a different treatment, and its size depends on the number of patients that were directly examined. The nodes are joined by lines with different thicknesses, which show whether there was a direct relationship between treatments, and the thickness is weighted according to the available direct evidence between them.

**Figure 2 cancers-15-05698-f002:**
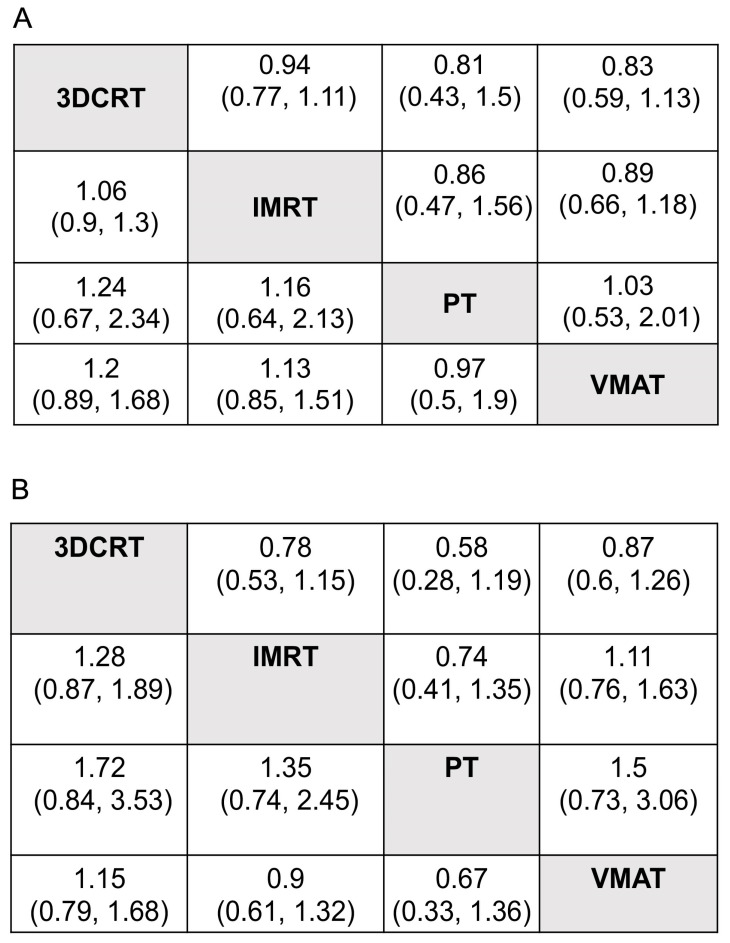
Pooled estimates for all possible treatment effects for each outcome. (**A**): Overall survival; (**B**): progression-free survival. The efficacy of different treatment regimens using HRs and corresponding 95% Cis. All results are displayed as the ratio of the x-axis versus the y-axis.

**Figure 3 cancers-15-05698-f003:**
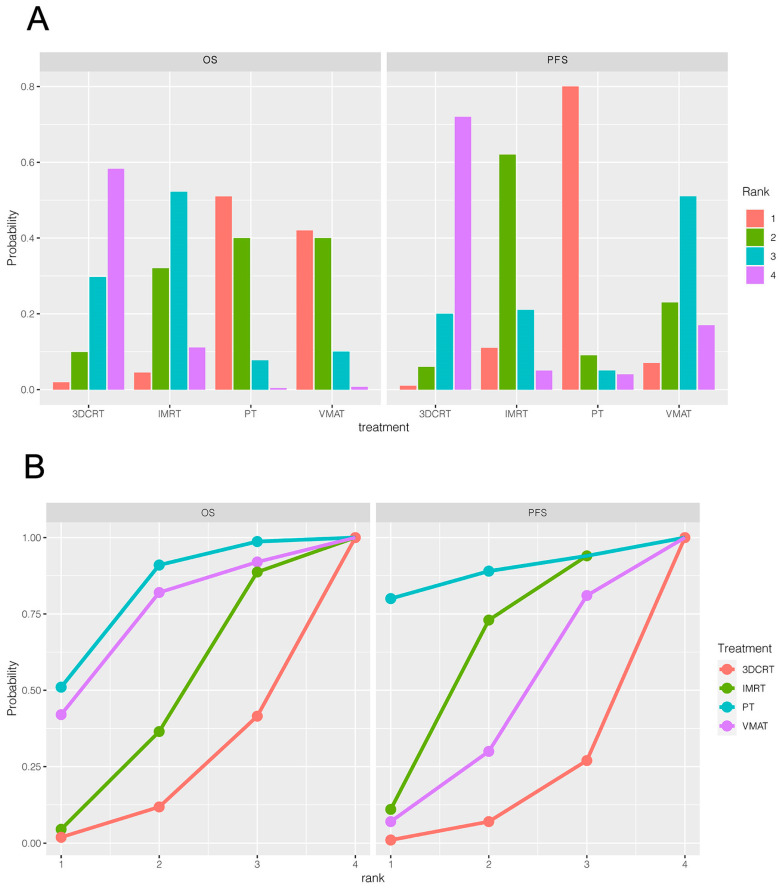
Rank of the efficacy and the SUCRA plot of different treatment regimens. (**A**): Rank of the efficacy of different treatment regimens; (**B**): cumulative rank plot with the surface under the cumulative ranking curve (SUCRA) of each intervention for effective outcomes.

**Table 1 cancers-15-05698-t001:** Overview of included studies.

Author/ Year	Design	Treatment Planning	Patients (n)	Median Age (y)	Extent of Resection n (%)	Radiation Treatment Planning	Median Follow-Up	OS (Median)	PFS (Median)
Brown 2021 [1]	P	PT	26	55.2	GTR: 19 (73.1%) STR: 5 (19.2%) Biopsy: 2 (7.7%)	GTV:T1 CTV:GTV+2 cm PTV:CTV+3-5 mm	48.7 months	24.5 months	8.9 months
		IMRT	41	52	GTR: 20 (48.8%) STR: 17 (41.5%) Biopsy: 4 (9.8%)			21.2 months	6.6 months
Chen 2013 [20]	R	3D-CRT	33	47	GTR/STR: 23 (69.7) Partial/biopsy: 10 (30.3)	GTV1: FLAIR or T2 CTV:GTV1+2 cm PTV:CTV+0.5 cm	13 months	NR^1^	NR^1^
		IMRT	21	47	GTR/STR: 11 (52.3) Partial/biopsy: 10 (47.7)			NR^1^	NR^1^
Navarria 2016 [21]	R	3D-CRT	167	53	GTR: 47 (28%) STR: 15 (9%) Partial: 58 (35%) Biopsy:47 (28%)	GTV: FLAIR or T1 CTV: GTV+10 mm PTV:CTV+3 mm	1.3 years	14.52 months	11.88 months
		VMAT	174	54	GTR: 68 (39%) STR: 26 (15%) Partial: 58 (33%) Biopsy: 22 (13%)			18.72 months	15.48 months
Thibouw 2018 [22]	R	3D-CRT	142	61	GTR: 21 (14.8%) STR: 30 (21.1%) Partial: 32 (22.5%) Biopsy: 59 (41.5%)	GTV:T1 CTV:GTV+15 mm PTV:CTV+3-5 mm	12.9 months	16.0 months	NR
		IMRT	78	62.5	GTR: 10 (12.8%) STR: 18 (23.1%) Partial: 26 (33.3%) Biopsy: 24 (30.8%)			13.4 months	NR
Huilgol 2018 [23]	R	IMRT	28	58.5	GTR: 7 (25%) STR: 17 (60.7%) Biopsy: 4 (14.3%)	NR	22.27 months	19.22 months	NR
		3D-CRT	18	54.11	GTR: 4 (22.2%) STR: 10 (55.6%) Biopsy: 4 (22.2%)			19.22 months	NR
Sheu 2019 [24]	R	VMAT	43	56	GTR: 53%	GTV:T1 CTV:GTV+2 cm PTV:CTV+3-5 mm	27.4 months	22.0 months	8.0 months
		IMRT	45	53	GTR: 60%			18.4 months	8.8 months

R: retrospective study; P: prospective study; GTR: gross total resection; STR: subtotal resection; NR: not reported; NR^1^: Not reported but estimated from published Kaplan–Meier curves. Radiotherapy with temozolomide was used in all studies.

**Table 2 cancers-15-05698-t002:** Surface under the cumulating ranking curve (SUCRA) data for all outcomes.

**A. SUCRA rankings for overall survival**	
PT	72.6
VMAT	66.5
IMRT	44.9
3DCRT	26.3
**B. SUCRA rankings for progression-free survival**	
PT	78.25
VMAT	42
IMRT	57
3DCRT	21.3

SUCRA = Surface under the cumulative ranking curve.

## Data Availability

The data presented in this study are available on request from the corresponding author.

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
