# Peer review of "Comparison of the Effectiveness of Radiotherapy with 3D-CRT, IMRT, VMAT and PT for Newly Diagnosed Glioblastoma: A Bayesian Network Meta-Analysis"

_cancers, 2023, doi:10.3390/cancers15235698_

Round 1
Reviewer 1 Report
Comments and Suggestions for Authors
The authors present a comparison of the effectiveness of radiotherapy with 3D-CRT, IMRT, VMAT and protons for newly diagnosed glioblastoma using a Bayesian network meta-analysis.
The issue has a high relevance for clinicians. Methods and results are described in detail. The authors discuss their findings adequately and they provide actual and appropriate references.
Comments by section:
Page 1, lines 36-37:
‘gross total resection, followed by postoperative RT and chemotherapy with Temozolomide (TMZ), resulting in the best survival outcomes so far’
The authors are encouraged to modify this passage, since Herrlinger et al. reported better outcomes with lomustine-temozolomide therapy in case of MGMT promoter methylation (10.1016/S0140-6736(18)31791-4).
Page 1 line 41:
Please check reference number 5. Here, the authors address prostate cancer, not glioblastoma.
Author Response
|
1. Summary |
|
|
We sincerely appreciate your dedication and the time you have invested in reviewing our manuscript. On behalf of all the co-authors, I extend our heartfelt thanks for your insightful and constructive feedback. Enclosed with this letter is a detailed, point-by-point response to your comments. It is our hope that our responses have adequately addressed your valuable suggestions and insights
|
|
|
2. Point-by-point response to Comments and Suggestions for Authors |
|
|
Comments 1: Page 1, lines 36-37: ‘gross total resection, followed by postoperative RT and chemotherapy with Temozolomide (TMZ), resulting in the best survival outcomes so far’ The authors are encouraged to modify this passage, since Herrlinger et al. reported better outcomes with lomustine-temozolomide therapy in case of MGMT promoter methylation (10.1016/S0140-6736(18)31791-4).
Response 1: We thank the reviewer for raising this important point. We have changed this sentence to 'The treatment of choice for newly diagnosed GBM patients is maximally safe total excision followed by postoperative radiotherapy (RT) and temozolomide (TMZ) chemotherapy.
|
|
|
|
|
|
Comments 2: Page 1 line 41: Please check reference number 5. Here, the authors address prostate cancer, not glioblastoma. |
|
|
Response 2: We regret and acknowledge this mistake pointed out by the reviewer. We corrected this mistake and deleted this reference.
|
|
Reviewer 2 Report
Comments and Suggestions for Authors Commens and Suggestions This is a well written and well analyzed article. The authors suggest the potential benefit of proton therapy over other modalities discussed in the manuscript for the treatment of GBM. As discussed in the manuscript, PT has been known to be effective in selective cases where tumors are in areas described as organs at risk but it's role for large diffusely infiltrating tumors has not been well established as this type of GBMs are highly infiltrative and it was felt that PT might not provide an adequate local control as the conventional radiation field is limited to 2-3 cm from the tumor boundary. In this article, it was noted that these 6 studies were subgrouped to GTR, STR and BX only. Were they subanalyzed based on the extent of surgery and if so, was there a difference in OS and PFS among these 3 subgroups and can it be shared or discussed in the manuscript? There were some typos, grammatical and other minor issues that can be edited or modified as discussed below. line 2: ..3D-CRT, IMRT, VMAT, protons: Please consider switching the word to Proton Therapy or PT to make it consistent with the rest of 3D-CRT, IMRT, VMAT line 12, 15 and others: pa-tient, ran-domized. Please try to change the wording format so that the entire word can be carried over to the next line without hyphenated for easier reading line 17, 24 and others: regimes. meaning of word "regime or regimes" is not certain but if felt to be appropriate, please consider switching the word “regime” to “regimen” or modality in the manuscript as it doesn’t seem to be the correct word and it probably needs to be changed for the American English readers. line 32, 37 and others: .. gliomas[1]. so far[3]. Please check the space between the end of the sentence and the reference as some of them are not spaced. line 57. 57: It is demanding to compare... Please consider rephrase the sentence such as Comparing the current....modern RT modalities is felt to be indicated or warranted in order to.. line 126 evidenced to evidence line 150 Table 1 IDH-1 Normal suggest to change to wild type rather than normal. line 170: sFigure? if typo, remove s. line 238 Pareek at al [36] between similar[34] and In an analysis of.. sentences. Please delete line 260: VMAT arises a lower cost for the health care system. Please rephrase like VMAT makes less financial burden to the health care system. line 260, 261: resulted an allowable cost of around $9,100 unlike IMRT.. Please consider to rephrase it like: resulted in a substantially lower cost of $9,100 compared to IMRT... line 280 GBM patient survival. Please adjust the double space gap between GBM patient to GBM patient line 303: .. craniopharyngioma PT to craniopharyngioma, PT. Please add comma line 313 chemo-radiotherapy [55; 56] . to chemo-radiotherapy [55; 56]. Please delete the space between [55;56] and period line 322 favourable to favorable for American English readers if felt to be indicated line 330 In some studies PT was…Please add comma between In some studies, PT was Comments on the Quality of English LanguageAs discussed in comment/suggestion section above, this manuscript needs some editing. Please consider a professional proof of English before submitting the final draft.
Author Response
|
1. Summary |
|
|
We sincerely appreciate your dedication and the time you have invested in reviewing our manuscript. On behalf of all the co-authors, I extend our heartfelt thanks for your insightful and constructive feedback. Enclosed with this letter is a detailed, point-by-point response to your comments. It is our hope that our responses have adequately addressed your valuable suggestions and insights
|
|
|
2. Point-by-point response to Comments and Suggestions for Authors |
|
|
Comments 1: In this article, it was noted that these 6 studies were subgrouped to GTR, STR and BX only. Were they subanalyzed based on the extent of surgery and if so, was there a difference in OS and PFS among these 3 subgroups and can it be shared or discussed in the manuscript?
Response 1: We thank the reviewer for raising this important point. This is a very interesting issue. Unfortunately, we were not able to obtain relevant information in these studies. We have tried to contact the authors of the paper, but unfortunately, they have not done any in-depth research into the surgical method. In fact, some studies have shown that GTR improves both OS and PFS compared to other surgical approaches. For instance, a meta-analysis revealed that patients who underwent GTR had a median OS of 20 months, significantly higher than those who underwent subtotal resection (STR). This supports the general understanding that more extensive tumor removal is associated with better survival outcomes in GBM patients. For STR, where a significant portion but not all of the tumor is removed, has been shown to result in decreased OS and PFS compared to GTR. For patients undergoing STR, the median OS was found to be around 12 months, which is notably lower than that for GTR. because STR often leaves residual tumors, which rarely undergo spontaneous regression, further impacting survival outcomes negatively. The option of biopsy alone generally results in the least favorable outcomes in terms of OS and PFS when compared to GTR or STR. |
|
|
|
|
|
Comments 2: There were some typos, grammatical and other minor issues that can be edited or modified as discussed below. line 2: ..3D-CRT, IMRT, VMAT, protons: Please consider switching the word to Proton Therapy or PT to make it consistent with the rest of 3D-CRT, IMRT, VMAT |
|
|
Response 2: We regret and acknowledge this mistake pointed out by the reviewer. We corrected this mistake and changed parts are marked in red
|
|
|
Comments 3: line 12, 15 and others: pa-tient, ran-domized. Please try to change the wording format so that the entire word can be carried over to the next line without hyphenated for easier reading Response3: We thank the reviewer for this important note. Unfortunately, as the magazine has already formatted the article, we are unable to correct this problem, even though we have made some adjustments. We will report this problem to the editor and hope that it can be resolved.
Comments 4: line 17, 24 and others: regimes. meaning of word "regime or regimes" is not certain but if felt to be appropriate, please consider switching the word “regime” to “regimen” or modality in the manuscript as it doesn’t seem to be the correct word and it probably needs to be changed for the American English readers. Response 4: We thank the reviewer for this important note. Now we changed regime or regimes to regimen.
Comments 5: line 32, 37 and others: .. gliomas[1]. so far[3]. Please check the space between the end of the sentence and the reference as some of them are not spaced. Response 5: We thank the reviewer for this important note. Now we corrected this mistake.
Comments 6: line 57. 57: It is demanding to compare... Please consider rephrase the sentence such as Comparing the current....modern RT modalities is felt to be indicated or warranted in order to.. Response 6: We thank the reviewer for this important note. Now we rephrase the sentence and changed parts are marked in red.
Comments 7: line 126 evidenced to evidence line Response 7: We thank the reviewer for this important note. Now we corrected this mistake.
Comments 8: 150 Table 1 IDH-1 Normal suggest to change to wild type rather than normal. Response 8: We thank the reviewer for this important note. Following comments from other reviewers, we have removed the information about IDH-1, but your suggestion is correct, IDH-1 should be wild type.
Comments 9: line 170: sFigure? if typo, remove s. Response 9: We thank the reviewer for this important note. Now we corrected this mistake.
Comments 10:line 238 Pareek at al [36] between similar[34] and In an analysis of.. sentences. Please delete Response 10: We thank the reviewer for this important note. Now we delete this sentence.
Comments 11. line 260: VMAT arises a lower cost for the health care system. Please rephrase like VMAT makes less financial burden to the health care system. Response 11: We thank the reviewer for this important note. Now we rephrase the sentence and changed parts are marked in red.
Comments 12. line 260, 261: resulted an allowable cost of around $9,100 unlike IMRT.. Please consider to rephrase it like: resulted in a substantially lower cost of $9,100 compared to IMRT... Response 12: We thank the reviewer for this important note. Now we rephrase the sentence and changed parts are marked in red.
Comments 13. line 280 GBM patient survival. Please adjust the double space gap between GBM patient to GBM patient Response 13: We thank the reviewer for this important note. Now we corrected this mistake
Comments 14. line 303: ..craniopharyngioma PT to craniopharyngioma, PT. Please add comma Response 14: We thank the reviewer for this important note. Now we add comma to here.
Comments 15. line 313 chemo-radiotherapy [55; 56] . to chemo-radiotherapy [55; 56]. Please delete the space between [55;56] and period Response 15: We thank the reviewer for this important note. Now we corrected this mistake
Comments 16. line 322 favourable to favorable for American English readers if felt to be indicated Response 16: We thank the reviewer for this important note. Now we corrected this mistake
Comments 17. line 330 In some studies PT was…Please add comma between In some studies, PT was Response 17: We thank the reviewer for this important note. Now we add comma to here.
|
|
Reviewer 3 Report
Comments and Suggestions for Authors
The subject the authors tackle, a comparison of past results of irradiation treatments for glioblastoma is an important one of great interest to oncologists and glioblastoma researchers. I recommentd publishing this work after much needed corrections are made.
Many of my suggestions can be ignored except 1] bad constructions on lines 54, 85, 93, 97, and many more , 2] the number of abbreviations in Abstract must be reduced to one or maximum two. 3] The many problems of Table 1 must be corrected, see comments below. 4] A general editing of language use is required to correct multiple contorted or outright incorrect English constructions. There are many more examples of poor or inscrutable language use needing correction than the ones mentioned below. Punctuation errors interfere with communication. These also must be corrected.
Line 12-13 technically is not a sentence but in scientific, medical writing and the context, I think this is fine, ok to use.
Lines 20-21. You say “optimal RT method” but would this not potentially change depending on location, potentially different molecular GB markers, pt age or other pt-specific variables ? As a personal guess, I think it is possible the optimal RT method will change depending on location within the brain and will change depending on residual GB tumor mass [which we cannot yet easily or reliably determine after gross total resection].
Since this paper will be read by non-specialists and PhD researchers as well as neuro oncologists, regarding 3DCRT, IMRT, VMAT and PT, I suggest a box with brief explanations as to what each one is, briefly how it is done and a note on availability, and advantages and disadvantages of each was would be. I recommend including Bragg peak in this explanatory box.
To limit abbreviations the authors could consider not using NMA, DIC, OAR, ON. I think these abbreviations are superfluous and needlessly complicate reading. This could be my idiosyncratic view but I think lowering number of abbreviations facilitates readers’ ease of understanding and lowers readers’ efforts required.
Line 54, slightly odd construction and potentially incorrect. Maintenance of QOL is indeed an important goal but as we all understand, maintaining QOL must be balanced against potential for cure or significant prolongation of life. I believe it is foremost the patient’s decision where the set point is on that spectrum. We can, and should, slant the spectrum according to our estimation of likelihoods but within that spectrum we present to pt, the pt decides. So I don’t agree “essential part of…”. Important consideration yes, but not essential. Balance is essential and pt determines where that balance sits. Do the authors agree ?
Re lines 58-60- Please rephrase. Say this simply and directly. Something like “The wide array of RT regimens available to treat post-resection GB can be difficult for pts and clinicians to evaluate. We designed this review to help in that decision making.” maybe ?
Line 85, bad construction [“ Studies were excluded as meeting the following criteria…”].
Say this simply and directly. “Studies meeting the following criteria were excluded:...”.
Or you could say “Studies were excluded if meeting the following criteria…”.
Line 93, there is no justification for using upper case T in temozolomide. Temozolomide is a generic drug name. The proprietary drug, Temodar, is a proper noun, so as such does begin with upper case T but temozolomide is a common noun so must begin with lower case t unless at beginning of sentence or in a title.
Line 97, unclear [Authors of included studies were contacted as important data were unclear or not reported.] Do you intend to say “if important data…” ? If yes, change. If no, rephrase.
Lines 116 to 136. I was unable to evaluate these lines due to my own limitations/ignorances.
Line 138. In subheading either all upper or all lower case.
I found Table 1 very difficult to follow on several grounds.
-
Unusual splitting up of words not by syllables, the crowded data presentation, irregular use of decimals, unnecessary columns et al.
-
The TMZ column is superfluous because all studies used it so that can be mentioned in the Table’s description. 3.
-
Age column is not meaningful IMO. Can you say “median age in these studies varied between 47 and 62 years old”.
-
Authors list study duration to the hundredth place sometimes. This is both silly and misleading.
-
If authors are not going to show results separately for each category then I recommend not specifying % in columns “IDH”, “extent of resection”,
-
Unclear what column “Radiation treatment planning” means or why it is shown.
-
“Median follow up” column is fine but showing the range is not helpful. Yes, the range will be important to someone who is interested in checking your results and independently evaluating the authors’ dataset, but just clutters the Table without benefit to most readers.
-
So I think these columns can be deleted with an increase in information delivery as consequence - Design, Median age, IDH, Radiation treatment planning, Concurrent systemic therapy, and the range part of column Median follow up.
-
IDH is not designated “normal”. Correct would be wild type or non-mutated but I don’t see either as meaningful in your data. IDH mutated or not is very important as a prognostic or treatment-related characteristic but is not important if PFS and OS results are not separately listed in results.
-
I would say the same for showing extent of resection in the study. Why show this if you don’t show results also divided into those same categories ?
Legend requires fixing. Line 152 example, language and punctuation errors. “ NR; Not report;”
Should read “NR, not reported; NR1, not reported but estimated from published Kaplan- Meier curves;...” etc
Figure 1 is not helpful. I would think old fashioned bar graphs would be stronger quicker to grasp presentation. Maybe I am missing a point the authors are trying to convey here ?
Line 181, “CrIs” ? what is that ? Do you mean confidence interval that you previously defined as CI ? Perhaps I didn't give it enough time to decipher, but I found the matrix shown in Fig. 2 inscrutable.
Again the fault could be mine, but I found Figure 3 also hard to follow. The X axis is Probability but the legend says is is SCURA. Can the authors find a more effective way to graphically show us their results ?
Line 214, PTV never defined.
Error in line 216. “ lowering normal-tissue constraints in the brainstem” what constraints ? I think you meant “ lowering dose delivered to the brainstem…” ?
Language error Line 224 “IMRT is generally recommended superiorly “ ? do the authors intend to say “IMRT is generally regarded as superior” ?
Line 245, re. “Toxicities”, do the authors mean off target brain damage as evidenced by deficits, or necrosis or what ?
Line 255, Sorry, I did not understand “Although VMAT improved survival rates, the difference between VMAT and IMRT was not significant from the perspective of clinical benefits. “.
I think in Discussion section mention should be made of pharmacological augmentation strategies during irradiation. An augmentation strategy for one mode may not be applicable to another mode. So far no pharmacological irradiation augmentation regimen has shown unequivocal benefit but multiple trials are in progress. This field is in active development and not covered in this paper but should one or another pharmacological irradiation strategy prove effective this would potentially change our conclusions.
The subject the authors tackle, a comparison of past results of irradiation treatments for glioblastoma is an important one of great interest to oncologists and glioblastoma researchers. I recommentd publishing this work after much needed corrections are made.
Many of my suggestions can be ignored except 1] bad constructions on lines 54, 85, 93, 97, and many more , 2] the number of abbreviations in Abstract must be reduced to one or maximum two. 3] The many problems of Table 1 must be corrected, see comments below. 4] A general editing of language use is required to correct multiple contorted or outright incorrect English constructions. There are many more examples of poor or inscrutable language use needing correction than the ones mentioned below. Punctuation errors interfere with communication. These also must be corrected.
Line 12-13 technically is not a sentence but in scientific, medical writing and the context, I think this is fine, ok to use.
Lines 20-21. You say “optimal RT method” but would this not potentially change depending on location, potentially different molecular GB markers, pt age or other pt-specific variables ? As a personal guess, I think it is possible the optimal RT method will change depending on location within the brain and will change depending on residual GB tumor mass [which we cannot yet easily or reliably determine after gross total resection].
Since this paper will be read by non-specialists and PhD researchers as well as neuro oncologists, regarding 3DCRT, IMRT, VMAT and PT, I suggest a box with brief explanations as to what each one is, briefly how it is done and a note on availability, and advantages and disadvantages of each was would be. I recommend including Bragg peak in this explanatory box.
To limit abbreviations the authors could consider not using NMA, DIC, OAR, ON. I think these abbreviations are superfluous and needlessly complicate reading. This could be my idiosyncratic view but I think lowering number of abbreviations facilitates readers’ ease of understanding and lowers readers’ efforts required.
Line 54, slightly odd construction and potentially incorrect. Maintenance of QOL is indeed an important goal but as we all understand, maintaining QOL must be balanced against potential for cure or significant prolongation of life. I believe it is foremost the patient’s decision where the set point is on that spectrum. We can, and should, slant the spectrum according to our estimation of likelihoods but within that spectrum we present to pt, the pt decides. So I don’t agree “essential part of…”. Important consideration yes, but not essential. Balance is essential and pt determines where that balance sits. Do the authors agree ?
Re lines 58-60- Please rephrase. Say this simply and directly. Something like “The wide array of RT regimens available to treat post-resection GB can be difficult for pts and clinicians to evaluate. We designed this review to help in that decision making.” maybe ?
Line 85, bad construction [“ Studies were excluded as meeting the following criteria…”].
Say this simply and directly. “Studies meeting the following criteria were excluded:...”.
Or you could say “Studies were excluded if meeting the following criteria…”.
Line 93, there is no justification for using upper case T in temozolomide. Temozolomide is a generic drug name. The proprietary drug, Temodar, is a proper noun, so as such does begin with upper case T but temozolomide is a common noun so must begin with lower case t unless at beginning of sentence or in a title.
Line 97, unclear [Authors of included studies were contacted as important data were unclear or not reported.] Do you intend to say “if important data…” ? If yes, change. If no, rephrase.
Lines 116 to 136. I was unable to evaluate these lines due to my own limitations/ignorances.
Line 138. In subheading either all upper or all lower case.
I found Table 1 very difficult to follow on several grounds.
-
Unusual splitting up of words not by syllables, the crowded data presentation, irregular use of decimals, unnecessary columns et al.
-
The TMZ column is superfluous because all studies used it so that can be mentioned in the Table’s description. 3.
-
Age column is not meaningful IMO. Can you say “median age in these studies varied between 47 and 62 years old”.
-
Authors list study duration to the hundredth place sometimes. This is both silly and misleading.
-
If authors are not going to show results separately for each category then I recommend not specifying % in columns “IDH”, “extent of resection”,
-
Unclear what column “Radiation treatment planning” means or why it is shown.
-
“Median follow up” column is fine but showing the range is not helpful. Yes, the range will be important to someone who is interested in checking your results and independently evaluating the authors’ dataset, but just clutters the Table without benefit to most readers.
-
So I think these columns can be deleted with an increase in information delivery as consequence - Design, Median age, IDH, Radiation treatment planning, Concurrent systemic therapy, and the range part of column Median follow up.
-
IDH is not designated “normal”. Correct would be wild type or non-mutated but I don’t see either as meaningful in your data. IDH mutated or not is very important as a prognostic or treatment-related characteristic but is not important if PFS and OS results are not separately listed in results.
-
I would say the same for showing extent of resection in the study. Why show this if you don’t show results also divided into those same categories ?
Legend requires fixing. Line 152 example, language and punctuation errors. “ NR; Not report;”
Should read “NR, not reported; NR1, not reported but estimated from published Kaplan- Meier curves;...” etc
Figure 1 is not helpful. I would think old fashioned bar graphs would be stronger quicker to grasp presentation. Maybe I am missing a point the authors are trying to convey here ?
Line 181, “CrIs” ? what is that ? Do you mean confidence interval that you previously defined as CI ? Perhaps I didn't give it enough time to decipher, but I found the matrix shown in Fig. 2 inscrutable.
Again the fault could be mine, but I found Figure 3 also hard to follow. The X axis is Probability but the legend says is is SCURA. Can the authors find a more effective way to graphically show us their results ?
Line 214, PTV never defined.
Error in line 216. “ lowering normal-tissue constraints in the brainstem” what constraints ? I think you meant “ lowering dose delivered to the brainstem…” ?
Language error Line 224 “IMRT is generally recommended superiorly “ ? do the authors intend to say “IMRT is generally regarded as superior” ?
Line 245, re. “Toxicities”, do the authors mean off target brain damage as evidenced by deficits, or necrosis or what ?
Line 255, Sorry, I did not understand “Although VMAT improved survival rates, the difference between VMAT and IMRT was not significant from the perspective of clinical benefits. “.
I think in Discussion section mention should be made of pharmacological augmentation strategies during irradiation. An augmentation strategy for one mode may not be applicable to another mode. So far no pharmacological irradiation augmentation regimen has shown unequivocal benefit but multiple trials are in progress. This field is in active development and not covered in this paper but should one or another pharmacological irradiation strategy prove effective this would potentially change our conclusions.
Author Response
|
1. Summary |
|
|
We sincerely appreciate your dedication and the time you have invested in reviewing our manuscript. On behalf of all the co-authors, I extend our heartfelt thanks for your insightful and constructive feedback. Enclosed with this letter is a detailed, point-by-point response to your comments. We've also made English grammar changes throughout the text. We hope that our responses have adequately addressed your valuable suggestions and insights.
|
|
|
2. Point-by-point response to Comments and Suggestions for Authors Comments 1: Many of my suggestions can be ignored except bad constructions on lines 54, 85, 93, 97, and many more , Response: We regret and acknowledge the error pointed out by the reviewer. We have corrected this error and rephrased the sentence, and the changed parts are marked in red.
Comments 2: the number of abbreviations in Abstract must be reduced to one or maximum two. Response: We thank the reviewer for this important note. Now we have only 2 abbreviations in our Abstract.
Comments 3: The many problems of Table 1 must be corrected, see comments below. Response: Thank you for your comments. We have modified Table 1 based on your suggestions.
Comments 4: A general editing of language use is required to correct multiple contorted or outright incorrect English constructions. There are many more examples of poor or inscrutable language use needing correction than the ones mentioned below. Punctuation errors interfere with communication. These also must be corrected.
Comments 5: Line 12-13 technically is not a sentence but in scientific, medical writing and the context, I think this is fine, ok to use. Response: We thank the reviewer for your comments. we rewrote this part and the information is presented in red font.
Comments 6: Lines 20-21. You say “optimal RT method” but would this not potentially change depending on location, potentially different molecular GB markers, pt age or other pt-specific variables ? As a personal guess, I think it is possible the optimal RT method will change depending on location within the brain and will change depending on residual GB tumor mass [which we cannot yet easily or reliably determine after gross total resection]. Response: We thank the reviewer for your comments. you are right, the final efficacy also depends on the location of the tumor in the brain and the residual tumor.we rewrote this part and the information is presented in red font.
Comments 7: Since this paper will be read by non-specialists and PhD researchers as well as neuro oncologists, regarding 3DCRT, IMRT, VMAT and PT, I suggest a box with brief explanations as to what each one is, briefly how it is done and a note on availability, and advantages and disadvantages of each was would be. I recommend including Bragg peak in this explanatory box. Response: We thank the reviewer for your comments. we add these information in our discussion and the information is presented in red font.
Comments 8: To limit abbreviations the authors could consider not using NMA, DIC, OAR, ON. I think these abbreviations are superfluous and needlessly complicate reading. This could be my idiosyncratic view but I think lowering number of abbreviations facilitates readers’ ease of understanding and lowers readers’ efforts required. Response: We thank the reviewer for your suggestion. We have reduced the number of abbreviations in the paper, including NMA, DIC, OAR, ON.
Comments 9: Line 54, slightly odd construction and potentially incorrect. Maintenance of QOL is indeed an important goal but as we all understand, maintaining QOL must be balanced against potential for cure or significant prolongation of life. I believe it is foremost the patient’s decision where the set point is on that spectrum. We can, and should, slant the spectrum according to our estimation of likelihoods but within that spectrum we present to pt, the pt decides. So I don’t agree “essential part of…”. Important consideration yes, but not essential. Balance is essential and pt determines where that balance sits. Do the authors agree ? Response: We regret and acknowledge the error pointed out by the reviewer. And you are right. This sentence is inappropriate. We have deleted it.
Comments 10: Re lines 58-60- Please rephrase. Say this simply and directly. Something like “The wide array of RT regimens available to treat post-resection GB can be difficult for pts and clinicians to evaluate. We designed this review to help in that decision making.” maybe ? Response: We thank the reviewer for this important note. Now we rephrase the sentence and changed parts are marked in red.
Comments 11: Line 85, bad construction [“ Studies were excluded as meeting the following criteria…”]. Say this simply and directly. “Studies meeting the following criteria were excluded:...”. Or you could say “Studies were excluded if meeting the following criteria…”. Response: We thank the reviewer for this important note. Now we rephrase the sentence and changed parts are marked in red.
Comments 12: Line 93, there is no justification for using upper case T in temozolomide. Temozolomide is a generic drug name. The proprietary drug, Temodar, is a proper noun, so as such does begin with upper case T but temozolomide is a common noun so must begin with lower case t unless at beginning of sentence or in a title. Response: We thank the reviewer for this important note. Now we change to temozolomide
Comments 13: Line 97, unclear [Authors of included studies were contacted as important data were unclear or not reported.] Do you intend to say “if important data…” ? If yes, change. If no, rephrase. Response: We thank the reviewer for this important note. Now we rephrase the sentence and changed parts are marked in red.
Comments 14: Lines 116 to 136. I was unable to evaluate these lines due to my own limitations/ignorances. Response: Our method is based on the study by Devji et al. We have also published another paper using this method before.
Devji T, Levine O, Neupane B, et al. Systemic therapy for previously untreated advanced BRAF-mutated melanoma: a systematic review and network meta-analysis of randomized clinical trials[J]. JAMA oncology, 2017, 3(3): 366-373.
Xu S, Sak A, Erol Y B. Network meta-analysis of first-line systemic treatment for patients with metastatic colorectal cancer[J]. Cancer Control, 2021, 28: 10732748211033497.
Comments 15: Line 138. In subheading either all upper or all lower case. Response: We thank the reviewer for this important note. This part is the editor's modification of our article, so we don't know what format is required. We will report this problem to the editor and hope that it can be resolved.
Comments 16: I found Table 1 very difficult to follow on several grounds.
1. Unusual splitting up of words not by syllables, the crowded data presentation, irregular use of decimals, unnecessary columns et al. Response: We thank the reviewer for this important note. We have organised the table according to your suggestions.
2. The TMZ column is superfluous because all studies used it so that can be mentioned in the Table’s description. 3. Response: We thank the reviewer for this important note. We remove this column from the table and mention .it in the description of the table.
3. Age column is not meaningful IMO. Can you say “median age in these studies varied between 47 and 62 years old”. Response: We thank the reviewer for this suggestion. After discussion, we decided to keep the age column. Because glioblastoma tends to have different prognoses and outcomes based on the age of the patient. Younger patients often have a better prognosis compared to older patients. This is due to various factors, including the general health of the patient, the ability to tolerate aggressive treatments, and potentially different biological characteristics of the tumor in different age groups. In addition, In clinical trials, understanding the median age of participants is essential for interpreting the results and applying them to the general population. Trials may need to be designed to specifically address the outcomes in different age groups.
4. Authors list study duration to the hundredth place sometimes. This is both silly and misleading. Response: We thank the reviewer for this important note. Now we changed to author/year.
5. If authors are not going to show results separately for each category then I recommend not specifying % in columns “IDH”, “extent of resection”, Response: We thank the reviewer for this important note. Following your comments, we have removed the information about IDH-1.
6. Unclear what column “Radiation treatment planning” means or why it is shown. Response: Thanks to the reviewer for raising this question. Radiation Treatment Planning: This is a meticulous process undertaken before radiation therapy where radiation oncologists and medical physicists plan the precise delivery of radiation to the cancerous area. The goal is to maximize the dose to the tumor while minimizing exposure to surrounding healthy tissues. Radiation Treatment Planning has very important reference significance for clinicians, so we retain this information
7. “Median follow up” column is fine but showing the range is not helpful. Yes, the range will be important to someone who is interested in checking your results and independently evaluating the authors’ dataset, but just clutters the Table without benefit to most readers. Response: We thank the reviewer for this important note. We removed the range.
8. So I think these columns can be deleted with an increase in information delivery as consequence - Design, Median age, IDH, Radiation treatment planning, Concurrent systemic therapy, and the range part of column Median follow up. Response: We thank the reviewer for this suggestion. We have organised the table according to your suggestions.
9. IDH is not designated “normal”. Correct would be wild type or non-mutated but I don’t see either as meaningful in your data. IDH mutated or not is very important as a prognostic or treatment-related characteristic but is not important if PFS and OS results are not separately listed in results. Response: We thank the reviewer for this important note. Following your comments, we have removed the information about IDH-1, but your suggestion is correct, IDH-1 should be wild type.
10. I would say the same for showing extent of resection in the study. Why show this if you don’t show results also divided into those same categories? Response: We thank the reviewer for this important note. The extent of resection is indeed an important consideration in the context of radiotherapy, particularly for brain tumors like glioblastoma. Firstly, the extent of tumor resection can significantly influence how radiotherapy is planned. After a more extensive resection, the remaining area that needs to be targeted by radiation may be smaller or differently shaped, affecting the radiation dose and its distribution. In addition, Studies have shown that a more extensive resection of the tumor can lead to better outcomes when followed by radiotherapy. This is because reducing the tumor burden may make the remaining cancer cells more susceptible to radiation. Besides, Understanding the extent of resection allows healthcare professionals to tailor the radiotherapy treatment to the individual needs of the patient, balancing the maximization of tumor control with the minimization of potential side effects. Therefore we present the relevant data in the table.
11. Legend requires fixing. Line 152 example, language and punctuation errors. “ NR; Not report;” Should read “NR, not reported; NR1, not reported but estimated from published Kaplan- Meier curves;...” etc Response: We regret and acknowledge the error pointed out by the reviewer. We have corrected this error and rephrased the sentence, and the changed parts are marked in red.
12. Figure 1 is not helpful. I would think old fashioned bar graphs would be stronger quicker to grasp presentation. Maybe I am missing a point the authors are trying to convey here ? Response: Thanks to the reviewer for raising this question. Figure 1 is our Network structure diagram. This Network structure diagram is graphical representations used in network meta-analysis (NMA) to visually depict the relationships and comparisons between various treatments or interventions being studied. These diagrams play a crucial role in conveying the complexity and interconnections of the treatments within the meta-analysis. Network structure diagrams provide a visual summary of all the evidence available, showing which treatments have been directly compared in studies. They help in identifying gaps in the evidence where direct comparisons between treatments have not been made. Besides, Clinicians and decision-makers can use these diagrams to understand the landscape of available treatments and their comparative effectiveness.
13. Line 181, “CrIs” ? what is that ? Do you mean confidence interval that you previously defined as CI ? Perhaps I didn't give it enough time to decipher, but I found the matrix shown in Fig. 2 inscrutable. Response: We regret and acknowledge the error pointed out by the reviewer. CrIs should be corresponding 95% credible intervals (CIs). We have corrected this error changed parts are marked in red.
14. Again the fault could be mine, but I found Figure 3 also hard to follow. The X axis is Probability but the legend says is is SCURA. Can the authors find a more effective way to graphically show us their results ? Response: Thanks to the reviewer for raising this question. Our method is based on the study by Devji et al. you can find their same eFigure 4. And other research used this SCURA plot as well. For each preventive strategy, the cumulative probability of being ranked 1st through 4th is displayed. The more the curve for a certain strategy is located toward the upper left corner, the higher its SUCRA value and the greater its performance.
Devji T, Levine O, Neupane B, et al. Systemic therapy for previously untreated advanced BRAF-mutated melanoma: a systematic review and network meta-analysis of randomized clinical trials[J]. JAMA oncology, 2017, 3(3): 366-373. Van den Eynde J, Cloet N, Van Lerberghe R, et al. Strategies to prevent acute kidney injury after pediatric cardiac surgery: a network meta-analysis[J]. Clinical Journal of the American Society of Nephrology, 2021, 16(10): 1480-1490.
15. Line 214, PTV never defined. Response: We thank the reviewer for this important note. Now we explain PTV in our article and changed parts are marked in red.
16. Error in line 216. “ lowering normal-tissue constraints in the brainstem” what constraints ? I think you meant “ lowering dose delivered to the brainstem…” ? Response: We thank the reviewer for this important note. Now we rephrase the sentence and changed parts are marked in red.
17. Language error Line 224 “IMRT is generally recommended superiorly “ ? do the authors intend to say “IMRT is generally regarded as superior” ? Response: We thank the reviewer for this important note. Now we rephrase the sentence and changed parts are marked in red.
18. Line 245, re. “Toxicities”, do the authors mean off target brain damage as evidenced by deficits, or necrosis or what ? Response: We thank the reviewer for this important note. Now we rephrase the sentence and changed parts are marked in red.
19. Line 255, Sorry, I did not understand “Although VMAT improved survival rates, the difference between VMAT and IMRT was not significant from the perspective of clinical benefits. “. Response: We regret and acknowledge the error pointed out by the reviewer. And you are right. This sentence is inappropriate. We have deleted it.
20. I think in Discussion section mention should be made of pharmacological augmentation strategies during irradiation. An augmentation strategy for one mode may not be applicable to another mode. So far no pharmacological irradiation augmentation regimen has shown unequivocal benefit but multiple trials are in progress. This field is in active development and not covered in this paper but should one or another pharmacological irradiation strategy prove effective this would potentially change our conclusions. Response: We thank the reviewer for this important note. We now include this part of the discussion and changed parts are marked in red.
|
|